# Effects of Boride Orientation and Si Content on High-Temperature Oxidation Resistance of Directionally Solidified Fe–B Alloys

**DOI:** 10.3390/ma15217819

**Published:** 2022-11-05

**Authors:** Pengjia Guo, Shengqiang Ma, Xuebin He, Ping Lv, Yang Luo, Junhong Jia, Xudong Cui, Liujie Xu, Jiandong Xing

**Affiliations:** 1State Key Laboratory for Mechanical Behavior of Materials, School of Materials Science and Engineering, Xi’an Jiaotong University, Xi’an 710049, China; 2Shaanxi Union Research Center of University and Enterprise for Zinc-Based New Materials, Xi’an 710049, China; 3College of Mechanical & Engineering, Shaanxi University of Science and Technology, Xi’an 710021, China; 4National Joint Engineering Research Center for Abrasion Control and Molding of Metal Materials, Henan University of Science and Technology, Luoyang 471000, China

**Keywords:** directionally solidified (DS) Fe–B alloys, boride orientation, static oxidation, anti-oxidation performance

## Abstract

In this work, the as-cast directionally solidified (DS) Fe–B alloys with various Si contents and different boride orientation were designed and fabricated, and the as-cast microstructures and static oxidation behaviors of the DS Fe–B alloys were investigated extensively. The as-cast microstructure of the DS Fe–B alloys consists of the well-oriented Fe_2_B columnar grains and α-Fe, which are strongly refined by Si addition. The oxidation interface of the scales in the DS Fe–B alloy with 3.50 wt.% Si demonstrates an obvious saw-tooth shaped structure and is embedded into the alternating distributed columnar layer structures of the DS Fe–B alloy with oriented Fe_2_B and α-Fe matrix, which is beneficial to improve the anti-peeling performance of the oxide film compared with lower amounts of Si addition in DS Fe–B alloys with oriented Fe_2_B [002] orientation parallel to the oxidation direction (i.e., oxidation diffusion direction, labeled as Fe_2_B_//_ sample). In the DS Fe–B alloys with oriented Fe_2_B [002] orientation vertical to the oxidation direction (i.e., labeled as Fe_2_B_⊥_ sample), due to the blocking and barrier effect of laminated-structure boride, Si is mainly enriched in the lower part of the oxide film to form a dense SiO_2_ thin layer adhered to layered boride. As a result, the internal SiO_2_ thin layer plays an obstructed and shielded role in oxidation of the substrate, which hinders the further internal diffusion of oxygen ions and improves the anti-oxidation performance of the Fe_2_B_⊥_ sample, making the average anti-oxidation performance better than that of the Fe_2_B_//_ sample.

## 1. Introduction

In the past few decades, many efforts have been made to improve the high-temperature corrosion-wear behaviors of metal materials in metal liquids during the liquid metal forming [1,2,3,4,5]. There is good evidence that Fe–B alloys have good corrosion resistance due to the formation of a large number of stable and continuous borides in the matrix [6,7,8,9,10]. Moreover, boron has a very limited solubility in iron to consequently form borides, while carbon is almost insoluble in borides to produce carbides or largely dissolve into the iron matrix. Accordingly, the unique microstructures and properties of Fe–B alloys can be easily regulated by wide-range adjustment of the boron and carbon contents to control the hard-phase and anti-corrosive Fe_2_B and the continuously distributed metal matrix, which achieves independent control of the wear-resistant phase and matrix structure. Therefore, this alloy has better hardenability, fracture toughness and wear resistance compared to other iron-based alloys and steels [11,12,13,14,15,16,17]. As the primary alloying element, the high-temperature corrosion behaviors of borides have been investigated widely. Yan et al. examine the high-temperature behavior of the boride layer of 45# carbon steel and found that no oxidation occurred before 730 °C and only slight oxidation of the boride layer between 730 and 930 °C [18]. Kim et al. have reported that the addition of boron can improve the cavitation erosion resistance of Fe–Based hard-facing alloys [19]. Furthermore, Klein and Wu et al. revealed that appropriate B content in Co- and Nb-based alloys leads to improved oxidation resistance and oxide layer adhesion [20,21]. It has been found that borides tend to be oxidized to form B_2_O_3_ with a melting point of 450 °C at high temperatures [22]. B_2_O_3_ is highly hygroscopic and forms volatile HBO_2_ by forming B–O–H bonds with H_2_O in the air [23,24]. In this case, the volatilization and fluidity of glassy state B_2_O_3_ may generate some pores in the matrix, which allows the rapid diffusion of O ions, resulting in poor high-temperature oxidation resistance of Fe–B alloys. Pozdniakov et al. found that the steel with a high boron content was an important material for the production of spent nuclear fuel storage with little uncontrollable hot deformation behavior, in which the stainless-steel benefits from the distinct features of boron-containing steels [25].

Although some of the effects of boron content on high-temperature oxidation resistance of Fe–B alloys has been studied, the research on the high-temperature oxidation resistance of the DS Fe–B alloys by Si addition still lacks an understanding of the performance of oxidation. At present, the role of Si in high-speed steel has attracted more attention to discover its beneficial effects, such as formation of carbides, grain refinement and solid solution strengthening [26,27,28,29]. Moreover, Ju et al. investigated that Fe-10Cr-1.5B-6Al-0.3C-0.8Mn-0.6Si alloys reporting that they exhibited superior oxidation resistance at 900 °C [30]. Studies have also shown that the SiO_2_ layer generated at the interface of the oxide film can hinder ion diffusion [31]. Lv et al. reported that B_2_O_3_–SiO_2_ oxides accumulate in the inner oxide layer during chemical friction under high-temperature conditions, which improved the oxidation behavior of Fe–B alloys [32]. However, the effect of Si content and boride orientation on the high-temperature oxidation resistance of DS Fe–B alloy still needs to be studied to uncover the underlying synergistic actions. In this work, as-cast DS Fe–B alloys with various Si contents and boride orientations were designed and fabricated, the as-cast microstructures and static oxidation behaviors were investigated extensively to illustrate the synergistic effect of Si and the orientation of borides in DS Fe–B alloy.

## 2. Materials and Methods

### 2.1. Sample Preparation

The chemical composition of the investigated Fe–B alloys containing Si is listed in Table 1. Four samples of the Fe–B alloy with different Si contents were prepared and denoted as A1, A2, A3, and A4 samples. The Fe–B alloys were melted in a 10 kg vacuum induction melting furnace with charged materials of pure iron, ferrosilicon and ferrochromium [33]. When all the charged materials were melted in the furnace, the preheated raw ferroboron was added into the furnace after being deoxidize with a small amount of pure aluminum. Once the alloy had melted at 1450–1480 °C, it was poured into the designed circulating water-cooled mold to solidify directional (DS) microstructures, i.e., obtaining Y-block ingots of the DS Fe–B alloy, as shown in Figure 1. The directional region near the chilled mold, where the sample with an average solidification rate of approximately 16 °C/s was measured by a thermocouple [34]. The oxidation samples were prepared for the oxidation direction parallel to the preferential growth direction of Fe_2_B (i.e., defined as Fe_2_B_//_ sample) and the oxidation direction perpendicular to the Fe_2_B preferential growth direction (defined as Fe_2_B_⊥_ sample). The DS samples with two oxidation manners, that is, the Fe_2_B_//_ sample and the Fe_2_B_⊥_ sample, were oxidized at an elevated temperature with the oxidation direction parallel to the Fe_2_B [002] orientation and vertical to the Fe_2_B [002] orientation, and positions of the oxidation samples were cut from the Y-shaped DS ingots with the dimensions of 15 mm × 10 mm × 3 mm, as schematically shown in Figure 2. In order to prevent other surfaces of the sample from oxidation which may affect the experimental results, a small brush to apply an Al_2_O_3_ coating was used to overlay the coating on to the non-oxidized side of the specimen.

### 2.2. Static Oxidation

The static oxidation experiment method followed the Chinese national standards (GB/T 13303–91) [35]. The static oxidation tests were carried out in a box-type resistance furnace in air at 1073 K (with ±3 K) for a total oxidation time of 100 h. [32]. The static oxidation test was maintained at 20 h, 40 h, 60 h, 80 h and 100 h. After the oxidation test, the samples were cooled down in the furnace and then the weight gains were measured at room temperature on an electronic balance with a precision of 0.01 mg. In order to ensure the spalling of the scales, the samples were put into a crucible and the weight gains of the spalling scale after different oxidation durations were measured and calculated together according to GB/T 13303–91. Three repeated tests were conducted to obtain the average weight gain. The equipment used for static oxidation experiments is illustrated in Figure 3.

### 2.3. Characterization

All the metallographic specimens were polished using a diamond polishing agent with diamond particles of 0.5 μm suspended in a polishing fluid, and then etched using a 4% HNO_3_ solution. Samples with transverse sections of oxidation were prepared and planted with chemical nickel-plating, and then mounted in resin. After grinding with SiC sandpaper of up to 1500#, and polished by flannelette, the transverse sections of the oxidation samples were slightly etched using a 4% HNO_3_ solution. The microstructures of the as-cast samples were examined by scanning electron microscopy (SEM, VEGAII, XMUINCA, TESCAN, Brno, Czech Republic) and X-ray diffraction (XRD, D/Max-2400X, Rigaku Corporation, Tokyo, Japan). The MDI Jade 6 was used to identify (MDI Jade 6, Materials Data Management Inc., Indianapolis, IN, USA) the phase constituents of the as-cast and oxidation samples of the DS Fe–B alloys. The as-cast and cross-sectional samples before and after oxidation were characterized using X-ray diffraction (XRD), and scanning electron microscopy (SEM) with energy dispersive spectroscopy (EDS) to investigate the microstructures and morphologies as well as the scale thickness.

## 3. Results and Discussion

### 3.1. As-Cast Microstructure

Figure 4 shows the scanning electron microstructure images of the test alloys in the as-cast condition with different Si contents. From Figure 4, it can be seen that all the test alloys formed a metallic matrix and eutectic boride [36,37,38]. Clearly, the microstructures of the DS Fe–B alloys containing various Si addition showed good orientation effects. The columnar gray Fe_2_B were arranged with long rods with a preferred growth orientation of columnar Fe_2_B [002] crystal orientation [7,34]. Furthermore, the gray-white α-Fe matrix was distributed among the lamellar structures of columnar Fe_2_B, exhibiting dual-phase oriented microstructures of the DS Fe–B alloys (Figure 4). Additionally, the lamellar space of the columnar Fe_2_B was greatly refined by Si addition. Figure 5 shows the XRD patterns of the as-cast DS Fe–B alloys with different Si contents. It can be seen that the as-cast microstructures of the longitudinal section of DS Fe–B alloy mainly comprised Fe_2_B (36-1332) indexed with lots of strong peaks as well as the presence of some α-Fe peaks (06-0696) [32,39]. Obviously, the increasing Si addition resulted in the vanishing peaks of some small diffraction angles and strengthened peaks of large diffraction angles for oriented Fe_2_B, which may be influenced by the various oriented crystal planes of Fe_2_B grain in its lateral grain plane (i.e., prismatic plane of Fe_2_B with strong faceted crystal growth). Meanwhile, with the increase in Si content, there was no new phase in the DS Fe–B alloys, which indicates that Si just dissolves into the matrix and refines the lamellar structures of α-Fe and Fe_2_B [34].

### 3.2. The Static Oxidation Results

Figure 6 shows the weight gains per unit surface area for all DS samples at 1073 K. From Figure 6, it can be seen that the weight gains of Fe_2_B_//_ samples increased with an increase in oxidation time. The weight gain of all alloys rapidly developed in the initial oxidation stage of 20 h, and subsequently began to slow down. Moreover, the oxidation weight gain per unit area displayed a decreasing trend with the increase in Si addition, and sample A4 showed a steady and slight oxidation behavior and also exhibited the lowest oxidation weight gain. Obviously, the addition of Si can promote the oxidation resistance of the Fe_2_B_//_ samples of DS Fe–B alloy.

However, in the Fe_2_B_⊥_ sample of the DS Fe–B alloy, the oxidation weight gains of A2, A3 and A4 of Fe_2_B_⊥_ sample showed lower values than that of the Fe_2_B_//_ samples except for sample A1. That is to say, with the increase in Si addition, the DS Fe–B alloy with Fe_2_B_⊥_ sample possessed better oxidation resistance, which may be attributed to the Fe_2_B orientation effects, such as the Fe_2_B barrier layer and its oxidation resistance at different crystal orientations. Obviously, Si content in the matrix and Fe_2_B crystal orientation may have a synergistic role to demonstrate various oxidation resistances. When the oxidation time reached 40 h, the high-temperature oxidation weight gain curve of sample A4 with a Si addition of 3.50 wt.% displayed very slowly climbing behavior both in the Fe_2_B_//_ and Fe_2_B_⊥_ samples, which indicates that the increase in Si content has a strong oxidation inhibitory effect. It is worth noting that the static high-temperature oxidation resistance of the Fe_2_B_⊥_ sample was significantly better than that of the Fe_2_B_//_ sample, especially in the 1.50 wt.% Si and 2.50 wt.% Si samples (i.e., A2 and A3), but the A4 sample with 3.50 wt.% Si was approximately equal with the same values in the Fe_2_B_//_ and Fe_2_B_⊥_ samples. Undoubtedly, there is an obvious synergistic effect between Si addition and Fe_2_B crystal orientation.

The oxidation rate can be calculated using Equation (1) [40]:(1)$$\;K_{p}={\left(\frac{\Delta w}{A}\right)}^{2}t^{-1}$$
where *K_p_* represents the oxidation rate (mg^2^·cm^−4^·s^−1^), Δ*W* is the weight gain (mg), and *A* and t are the surface area (cm^2^) of the samples and oxidation time (s), respectively. After sample A4 was oxidized for 100 h, the oxidation rate of the Fe_2_B_⊥_ sample was 0.115 mg^2^·cm^−4^·s^−1^, which was better than that of the Fe_2_B_//_ sample with an oxidation rate of 0.152 mg^2^·cm^−4^·s^−1^.

### 3.3. Surface Morphology of Oxide Scales

The Fe_2_B_//_ oxidation surface morphologies of the oxide scales formed at 1073 K for 100 h are shown in Figure 7. It was observed that the oxide film on the surface of the DS Fe–B alloy without Si was relatively loose and rough with some holes and cracks, and more parts of the surface were cracked and wrinkled, which indicates that the oxide film was very easy to crack and peel off from the substrate (Figure 7a). From Figure 7b–d, with the increase in Si content, the oxide film of the DS Fe–B alloy became fairly compact, uniform and smooth. Furthermore, the emerging cracks and holes on the scales were gradually filled and vanished. Obviously, a large number of cracks and holes formed on the surface of sample A1 may provide fast channels for the diffusion of oxygen ions and thus promote the constant oxidation process [5]. With an increase in Si content, the inside of the oxide film became denser, preventing the outward diffusion of the internal matrix elements, and the surface became flatter. From Figure 8, the surface oxides of sample A1 were composed of Fe_2_O_3_, M_3_O_4_ (M = Fe, Cr) [41]. It is interesting to note that a small amount of SiO_2_ was obtained on the surface of the oxide film, which indicates that part of the Si may segregate at the substrate–scale interface and react with oxygen ions to form the inner SiO_2_ layer.

### 3.4. The Cross-Sectional Analysis of Oxide Scales

The cross-section morphologies for the Fe_2_B_//_ samples oxidized for 60 h at 1073 K are shown in Figure 9. From Figure 9a, it can be seen that the oxide film of sample A1 without Si addition was thicker, and the average oxide film thickness reached 126.035 μm. Furthermore, the overall oxide film was relatively porous and cracked, and there existed a large cleavage between the scales and substrate. Obviously, the combination of scale–substrate interface showed the internal bonding properties of the oxide films (e.g., cohesion of the scales, compactness of oxide film, etc.). However, when the Si content in the DS Fe–B alloy reached 1.50 wt.% and 2.50 wt.%, the thickness of the oxide film gradually decreased, and the oxide film became compact and continuous, as shown in Figure 9b,c (e.g., A2 and A3 alloys). As shown in Figure 9d, the oxide film was arranged in a zigzag pattern at the scale–substrate interface, which was deeply embedded and planted into the columnar matrix and was well pinned with the boride to form a packaged structure of an inner SiO_2_ film wrapped in columnar Fe_2_B grains. This means that oriented Fe_2_B at the inner interface can be pinned by internal growth SiO_2_ to generate an interlocking interface between oriented Fe_2_B and internally grown SiO_2_ so as to enhance the contact area of the scales and substrate as well as a pinning effect of the inner interface bulge or obstacle thus inhibiting the spallation of scales (as shown in Figure 9d for the A4 alloy). Clearly, the average thickness of cross-sectional oxide film decreased to 31.487 μm with the increase in Si content of 3.50 wt.%, as shown in Figure 9e. The A4 sample had the lowest oxidation rate of the Fe_2_B_//_ samples. Thus, this implied that the higher the silicon content of the matrix, the better the oxidation resistance of the DS alloy with Fe_2_B_//_ sample. There is no doubt that the addition of Si that forms an inner SiO_2_ film hinders the inner diffusion of oxygen ions into the substrate and reduces the further growth and spallation of the oxide films, thus improving the high-temperature oxidation resistance of the DS alloy [31].

Figure 10 shows the BSE micrographs of the cross-section of the Fe_2_B_⊥_ samples oxidized for 60 h at 1073 K. It can be seen from Figure 10a, that the thickness of the oxide film for sample A1 without Si addition was larger than others. From Figure 10c,d, it can be seen that the oxide films of the A3 and A4 samples were relatively flat and uniform, while the boride [002] orientation and oxidation interface of the oxide film are distributed in parallel (i.e., Fe_2_B [002] orientation vertical to the oxidation direction). It is clear that in the A4 sample, with Fe_2_B [002] orientation vertical to the oxidation interface (i.e., Fe_2_B_⊥_ sample with 3.50 wt.% Si addition), that some oxides formed by by-passing the boride with an emerging oxidation of the phase boundaries (i.e., α/Fe_2_B boundaries) appearing obliquely along the weak positions of the oriented Fe_2_B owing to the oxidation manner (i.e., Fe_2_B [002] orientation vertical to the oxidation direction) and probable Si segregation in these areas. By contrast, the average oxide film thickness of the Fe_2_B_⊥_ samples was smaller than that of the Fe_2_B_//_ samples, and the two oxidation manners of the Fe_2_B_//_ and Fe_2_B_⊥_ samples showed similar oxidation features with the increase in Si content in the DS Fe–B alloys (i.e., decrease in weight gains for both Fe_2_B_//_ and Fe_2_B_⊥_ samples with the increase in Si addition).

Figure 11 shows the high-magnification SEM morphology of the Fe_2_B_//_ and Fe_2_B_⊥_ samples of the DS A4 alloy oxidized at 1073 K for 60 h. From Figure 11a, the oxide film in the Fe_2_B_//_ sample was arranged in a zigzag pattern, and the internal oxidation was more serious owing to the insufficient synergy of oxidation between the matrix and oriented Fe_2_B in the DS Fe–B alloy despite more Si addition. However, the oriented boride and the oxide film were embedded into each other at their interfaces, which increased the bonding force and inhibited the spallation of scales. Meanwhile, it can be seen from Figure 11b that, the oriented boride was obliquely embedded into the oxide film and mainly played a barrier role in preventing the further oxidation and spallation of scales in the Fe_2_B_⊥_ sample. Evidently, the Fe_2_B_⊥_ sample dominated in oxidation resistance and possessed a better oxidation performance than the Fe_2_B_//_ sample, which was attributed to the lack of synergy between the matrix and oriented Fe_2_B in the Fe_2_B_//_ sample and the strong and beneficial barrier and inhibition of oxidation in the Fe_2_B_⊥_ sample to hinder the diffusion of oxygen ions and spallation of scales [42].

Figure 12 shows the cross-sectional line scanning analysis of the Fe_2_B_//_ sample of the A4 alloy oxidized for 60 h at 1073 K. The thickness of the oxide film was about 35 μm. From Figure 8 and Figure 12b, the oxide film was composed Fe_2_O_3_, Fe_3_O_4_ and SiO_2_ oxides. Fe_2_O_3_ and Fe_3_O_4_ mainly exist in the outer oxide layer, and an obvious internal oxidation is found at the boundary of substrate and oxide film (black zone at the bottom of the oxide film), it can be found that the inner oxide layer is a SiO_2_ layer. This SiO_2_ oxide scale is not only compact and continuous, but also presents strong and perfect adhesion to the metallic substrate [31,33], which may well hinder further O^2-^ diffusion into the matrix [31,33].

Figure 13 shows the EDS mapping of a cross-section of the Fe_2_B_⊥_ sample in the A4 alloy oxidized for 60 h at 1073 K. It is clear that the O element was uniformly distributed in the whole oxide film. However, due to the small diffusion coefficient of Si, it was concentrated in the lower part of the oxide film to form a thin layer of SiO_2_, particularly at the inner interface, which infers that the internal oxidation of the Si^+4^ cation occurs at the inner interface [5,43]. Si was sparsely distributed in the upper part of the oxide film, while Fe was distributed in the whole oxide film and can react with O^2−^ ions to form Fe_2_O_3_ and Fe_3_O_4_. Compared with the Fe_2_B_//_ oriented sample, Si in the Fe_2_B_⊥_ oriented sample was more concentrated and enriched at the bottom of the oxide film.

### 3.5. Mechanism of High-Temperature Oxidation Resistance of Directionally Solidified Fe–B Alloys with Different Orientations

According to the oxidation rate, the oxidation characteristics of DS Fe–B alloys are sensitive to the Si content and boride orientation. The oxidation results indicate that scale grows by a diffusion-controlled process. Since the diffusion coefficient of Si is much smaller than that of Fe, Si will tend to produce an internal oxide layer at the interface between the oxide film and the substrate [43,44]. The schematic diagram of the high-temperature oxidation resistance mechanism of DS Fe–B alloys with different boride orientations and Si content is shown in Figure 14. From Figure 14a, in the Fe_2_B_//_ sample, the Fe_2_B hard-phase in the matrix grew in a directional columnar arrangement and was embedded in the oxide film, and the upper part of the oxide film was mainly composed of Fe_2_O_3_ and Fe_3_O_4_. However, due to the voids and cracks in the oxide film, the metal cations could easily diffuse out and partly distribute into the oxide film during the oxidation. Additionally, voids and cracks could be used as a channel allowing O^2−^ ions to rapidly enter into the matrix of the DS alloy through this channel to form scales [23]. Since the free energy of Si oxide is lower than that of Fe oxide, it can be preferentially oxidized, and the inwardly diffused O^2−^ and Si^4+^ can form a significant SiO_2_ inner oxide layer [43,44,45]. The inwardly protruding SiO_2_ thin layer and the oriented Fe_2_B embedded into the oxide film pin each other increasing the bonding force of the scales, which can improve the anti-peeling performance of the oxide film. The formed SiO_2_ thin layer can further impede the inward diffusion of O^2−^ and improve the high-temperature oxidation resistance of the DS alloy [31].

Different from Figure 14b, in the Fe_2_B_⊥_ sample the Fe_2_B [002] orientation was parallel to the oxidation direction and was partly embedded into the oxide film obliquely. Due to the blocking effect of borides, Si^4+^ cannot continue to diffuse outward through the inner interface and is mainly concentrated and enriched at the bottom of the oxide film to form a SiO_2_ oxide layer, which depends on the O^2−^ inner-diffusion controlled process. Therefore, the overall anti-oxidation performance of the Fe_2_B_⊥_ sample was better than that of the Fe_2_B_//_ sample owing to the combined effects of Si content and Fe_2_B orientation.

## 4. Conclusions

In this work, the as-cast DS Fe–B alloys with various Si contents and different boride orientations were designed and fabricated, and their as-cast microstructures and static oxidation behaviors were investigated extensively. The main conclusions can be summarized as follows:(1)The as-cast DS Fe–B alloys consist of columnar Fe_2_B and α-Fe. The Si-containing microstructures of the DS Fe–B alloys show good orientation effects. The columnar gray Fe_2_B borides are arranged in long rods with Fe_2_B [002] crystal orientations as the preferred growth direction, and the gray-white α-Fe matrix is distributed among the laminated structure of Fe_2_B grains to display a dual-phase oriented microstructures of the DS Fe–B alloys.(2)When the preferred growth direction of Fe_2_B [002] orientation is parallel to the oxidation direction (marked as Fe_2_B_//_ sample), the oxidation weight gain decreases with the increase in Si content, and reaches the lowest value at 3.50 wt.% Si addition, which is attributed to the presence of a certain amount of an internal SiO_2_ film with obvious sawtooth-shaped structure except for outermost layer of Fe_2_O_3_, M_3_O_4_ (M = Fe, Cr) scales, which is embedded into the matrix and exhibits a beneficial role to improve the anti-peeling performance of the oxide film.(3)In the high-temperature static oxide film of the Fe_2_B_⊥_ sample (i.e., Fe_2_B [002] orientation perpendicular to the oxidation direction), Si is mainly concentrated and enriched in the lower part of the oxide film to form a SiO_2_ thin layer. The Si-rich internal oxidation layer and the blocking effect of oriented boride can effectively hinder further internal diffusion of oxygen ions and improve the anti-oxidation performance, which shows better oxidation resistance than that of the Fe_2_B_//_ sample.

## Figures and Tables

**Figure 1 materials-15-07819-f001:**
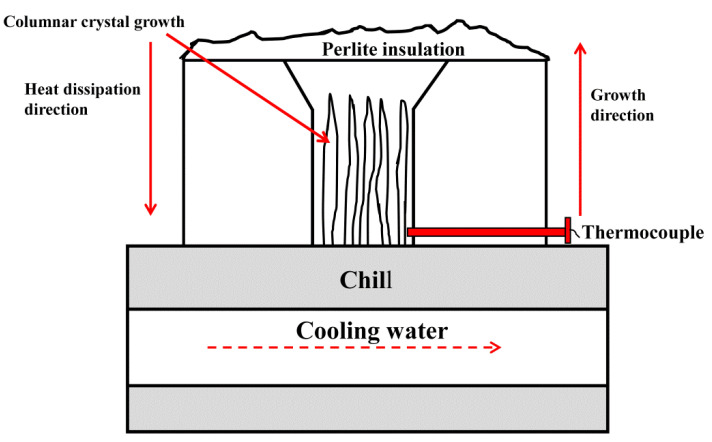
The schematic device of DS (directionally solidified) cast Fe–B alloy.

**Figure 2 materials-15-07819-f002:**
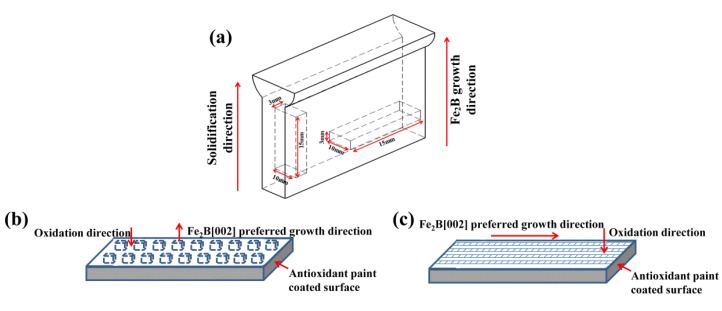
Schematic diagram of different oriented boride oxide samples: (**a**) Schematic diagram of sample position for high-temperature oxidation; (**b**) Fe_2_B_//_ samples; (**c**) Fe_2_B_⊥_ samples.

**Figure 3 materials-15-07819-f003:**
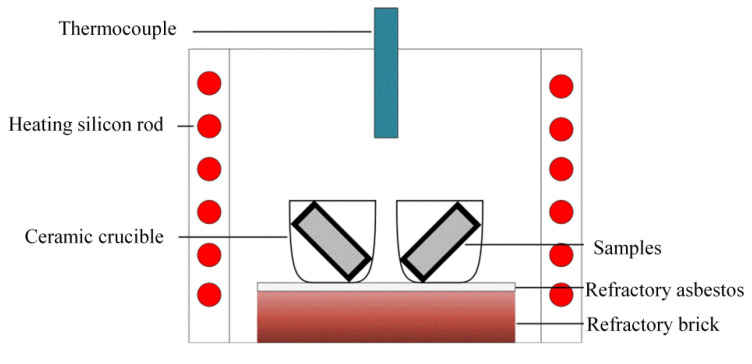
Schematic diagram of the static oxidation experiment apparatus.

**Figure 4 materials-15-07819-f004:**
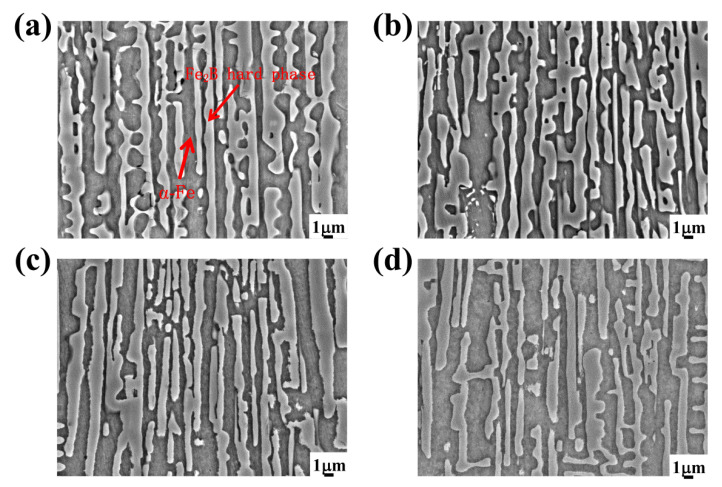
SEM micrographs of the as-cast microstructures: (**a**) A1, (**b**) A2, (**c**) A3, (**d**) A4.

**Figure 5 materials-15-07819-f005:**
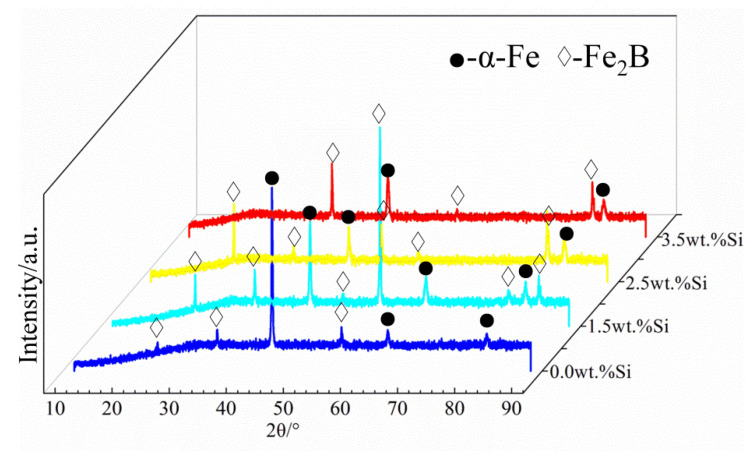
X-ray diffraction patterns of the as-cast DS Fe–B alloys (i.e., longitudinal section).

**Figure 6 materials-15-07819-f006:**
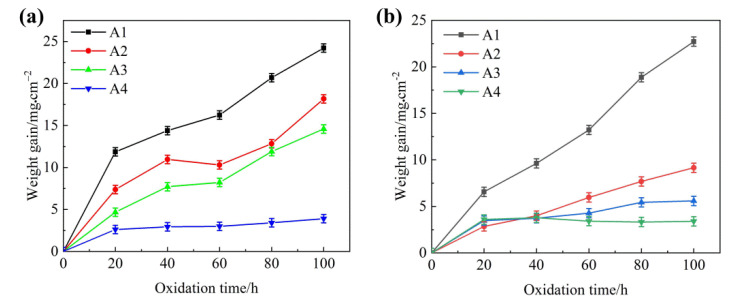
Weight gains per unit surface area as a function of oxidation time at 800 °C: (**a**) Fe_2_B_//_ samples with various Si addition, (**b**) Fe_2_B_⊥_ samples with various Si addition.

**Figure 7 materials-15-07819-f007:**
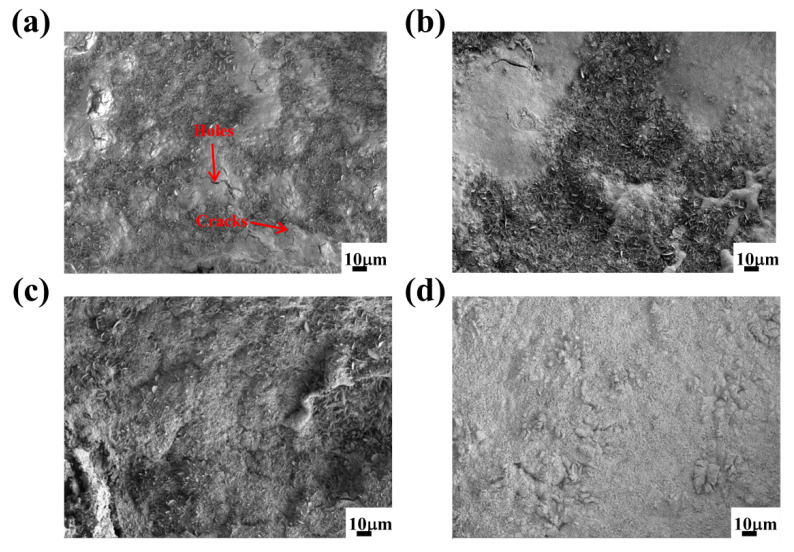
SEM micrographs of the Fe_2_B_//_ sample surface of the scale formed by 100 h oxidation at 1073 K: (**a**) alloy A1, (**b**) alloy A2, (**c**) alloy A3, (**d**) alloy A4.

**Figure 8 materials-15-07819-f008:**
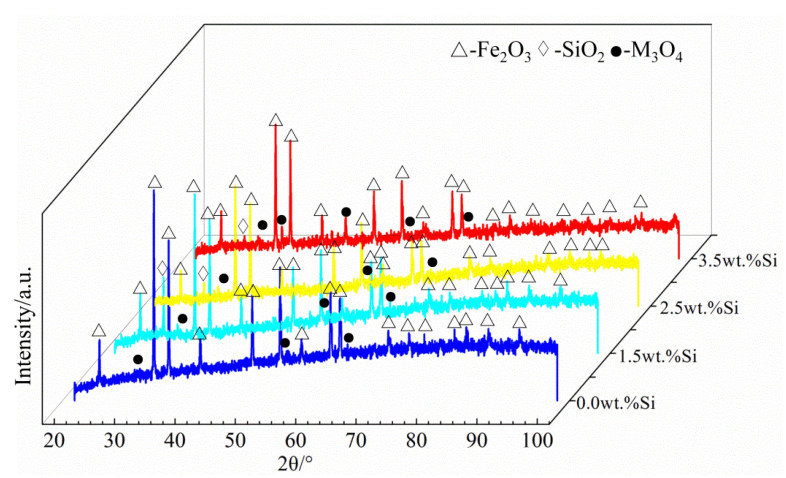
XRD on the surface of Fe_2_B_//_ sample oxide film after oxidation for 100 h.

**Figure 9 materials-15-07819-f009:**
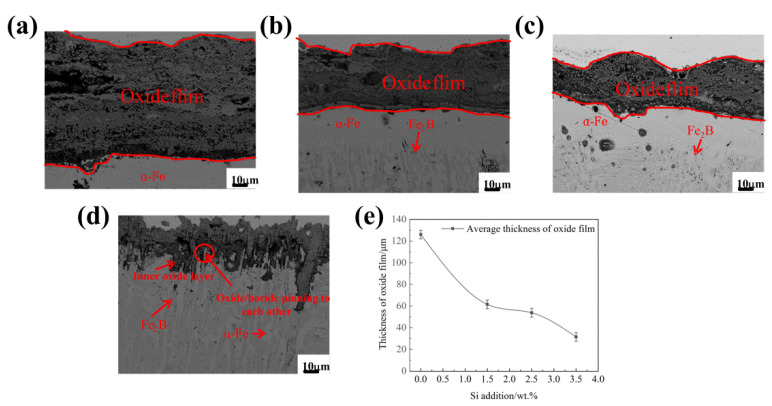
The BSE micrographs of the cross-section of Fe_2_B_//_ samples oxidized for 60 h at 1073 K: (**a**) alloy A1, (**b**) alloy A2, (**c**) alloy A3, (**d**) alloy A4, (**e**) the thickness of cross-sectional oxide film.

**Figure 10 materials-15-07819-f010:**
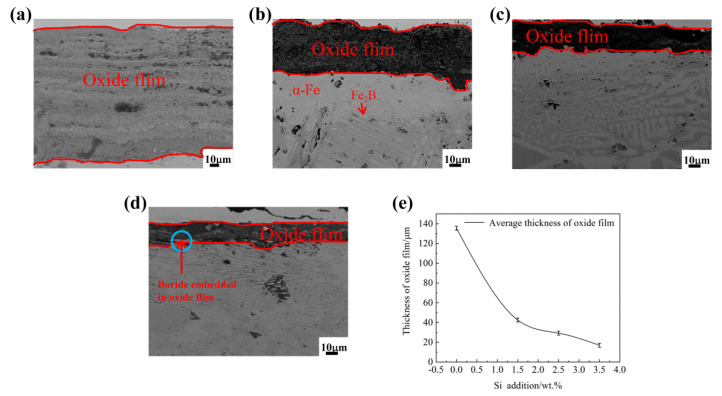
The BSE micrographs of the cross-section of Fe_2_B_⊥_ samples oxidized for 60 h at 1073 K: (**a**) alloy A1, (**b**) alloy A2, (**c**) alloy A3, (**d**) alloy A4, (**e**) the thickness of cross-sectional oxide film.

**Figure 11 materials-15-07819-f011:**
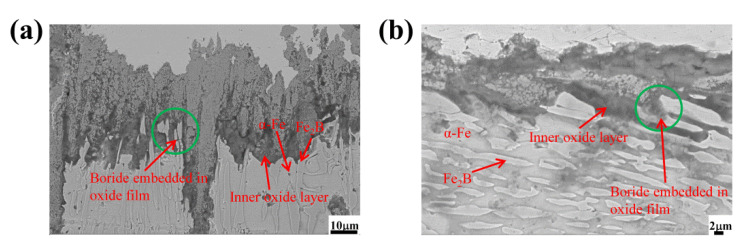
High-magnification SEM morphology of the A4 sample oxidized for 60 h at 1073 K: (**a**) Fe_2_B_//_ sample, (**b**) Fe_2_B_⊥_ sample.

**Figure 12 materials-15-07819-f012:**
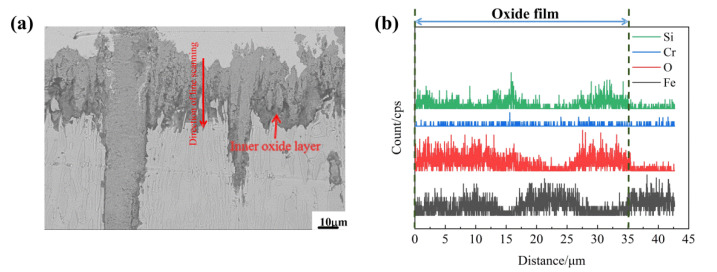
Line scanning analysis of a cross-section of the Fe_2_B_//_ A4 sample oxidized for 60 h at 1073 K: (**a**) diagram of line scan, (**b**) curve change diagram of each element of the line scan.

**Figure 13 materials-15-07819-f013:**
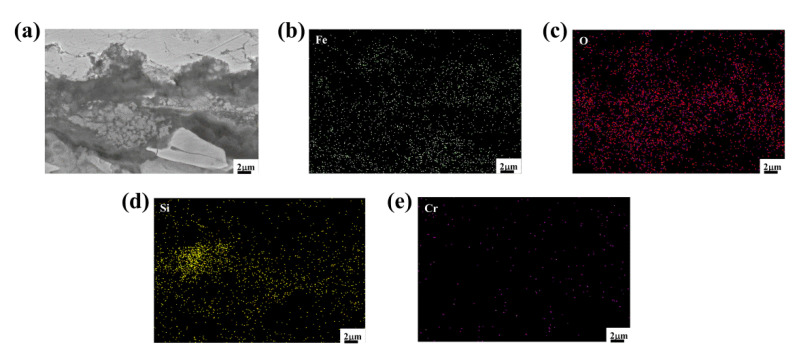
The EDS mapping of a cross-section of the Fe_2_B_⊥_ A4 sample oxidized for 60 h at 1073 K: (**a**) micrograph of surface scanning, (**b**) element distribution of Fe, (**c**) element distribution of O, (**d**) element distribution of Si, (**e**) element distribution of Cr.

**Figure 14 materials-15-07819-f014:**
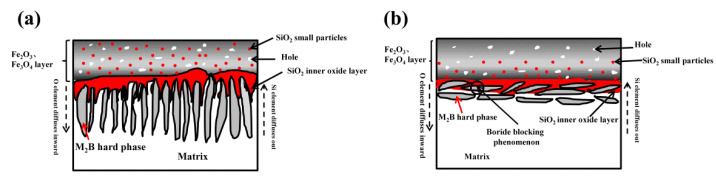
Schematic diagrams of high-temperature oxidation mechanisms: (**a**) Fe_2_B_//_ A4 sample, (**b**) Fe_2_B_⊥_ A4 sample.

**Table 1 materials-15-07819-t001:** Chemical compositions of cast DS Fe–B alloy by spark emission spectrometer (wt.%).

Samples	B	Cr	Si	C	Fe
A1	3.51	0.52	0.00	0.11	Bal
A2	3.51	0.50	1.50	0.10	Bal
A3	3.49	0.50	2.50	0.12	Bal
A4	3.50	0.51	3.50	0.11	Bal

## Data Availability

Data sharing is not applicable to this article.

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
