# Peer review of "Effects of Boride Orientation and Si Content on High-Temperature Oxidation Resistance of Directionally Solidified Fe–B Alloys"

_materials, 2022, doi:10.3390/ma15217819_

Round 1

Reviewer 1 Report

The authors of the paper “Effects of boride orientation and Si content on high temperature oxidation resistance of directionally solidified Fe-B alloys” have investigated the high-temperature corrosion properties of Fe-B-Si alloys. The authors have made a good experimental work. The authors described the technology of the material production and have investigated the microstructure of the alloys. However, some points in the manuscript are questionable. The manuscript may be accepted for publication after revision accordingly following comments:

1.                  In the introduction part the authors should describe a potential application of their materials. It is also recommended to describe the application of the high boron steels, such as cutting tools, nuclear application, etc. (please, consider works of Pozdniakov et. al).

2.                  What does mean phase Fe-Cr in the XRD-patterns? The amount of the Cr is too small to form any phases in the microstructure except Fe-based solid solution.

3.                  The phase Fe23B6 is absent in the binary phase diagram. What is the structure of this phase?

Reviewer 2 Report

Dear Authors

Lines 41-43. There is good evidence that Fe-B alloys have good corrosion resistance due to the formation of a large number of stable and continuous borides in the matrix [6-10]. Please, explain why You say so. Borides improve wear resistance, but corrosion resistance is improved if they are alloyed with significant amounts of chromium and/or nickel. 

Lines 84-86. Please, specify the cooling rate.

Fig. 2. Letters and digits are too small and cannot be read. Please, enlarge them. 

Fig. 4. All four microstructures here should be presented and compared.

Fig. 5. Please, make horizontal axis sale marks through 10 degrees, as on Fig. 8. This makes the picture easier to recognize. And change the XRD spectra to 3D representation because they overlap and hide the portions of each other. Please, specify the software and database used for the phase identification.

In section 2.2, please, disclose the type of antioxidant paint used. How did You apply it to the surface? How this could affect the test results?

Line 104. Did You use pure oxygen for the oxidation test? Or was it the air?

Line 123-124. Did You study the cross-sections of oxidized alloys? Not the oxidized surface?

Line 123. Please, specify, did You use diamond or alumina for polishing? What was the minimum grain size in the polishing suspension?

Line 151-154. Actually, Si is well-known as a steel deoxidizer, oxidation protector, and a self-fluxing additive. What is the novelty of Your discovery?

Line 195-198. This statement requires strong proof. Silicon oxide may be present as separate particles, not obviously as a layer.

Fig. 7. Please, enlarge the micrographs. They are now too small.

After Fig. 7, please, place the same 4 micrographs of Fe2B alloys to compare them with Fig. 7.

Fig. 9 and 0. Put here the same number (better all 4) of samples containing identical amounts of Si. The oxide film thickness graph in Fig. 10 looks better than in Fig. 9. For this figure, why did You make the ross-sections after 60h, but not after the end of the test, 100h?

Fig. 11. How long was the test? 100h? The microstructure in Fig. 11 b does not look like Fe2Bsample. The borides are diagonal to the surface. Please, repeat the test with proper grains orientation.

In the manuscript, many statements about oxide layer structure (especially about the silicon oxide layer) are rather speculation but not a matter of fact. In Fig. 13d silicon is also present at the surface of the oxide film, and has no a film distribution manner. I think, that You need to use more sensitive equipment (Zeiss Gemmini 500 or Thermofisher Scios dual beam) to prove this fact. 

Actually, oxidation - is a "scalar", and it has no direction. I think it is better to speak about boride grains: "borides parallel to oxidation surface, borides normal to the oxidation surface". Or "Oxygen diffusion is parallel/across to boride grains". 

Did You measure the surface area of borides when they are normal/parallel to the surface? Are these ratios identical for both samples? Due to the dissimilar oxidation resistance of iron and borides, this fat may significantly affect weight gain.

Please, do hard work to improve Your English. Many sentences (some of them are highlighted in yellow) are unclear and hard to understand

Round 2

Reviewer 2 Report

Dear Authors!

Thank You for the work done!